# Hepatocyte Growth Factor-mediated satellite cells niche perturbation promotes development of distinct sarcoma subtypes

Deborah Morena[1,2], Nicola Maestro[1,2†], Francesca Bersani[1,2‡], Paolo Emanuele Forni[1,2§], Marcello Francesco Lingua[1,2], Valentina Foglizzo[1,2¶], Petar Šćepanović[1,2**], Silvia Miretti[3], Alessandro Morotti[4], Jack F Shern[5], Javed Khan[5], Ugo Ala[6], Paolo Provero[6], Valentina Sala[1], Tiziana Crepaldi[1], Patrizia Gasparini[7], Michela Casanova[7], Andrea Ferrari[7], Gabriella Sozzi[7], Roberto Chiarle[2,6,8], Carola Ponzetto[1,2], Riccardo Taulli[1,2*]

[1]Department of Oncology, University of Turin, Turin, Italy; [2]CeRMS, Center for Experimental Research and Medical Studies, Turin, Italy; [3]Department of Veterinary Science, University of Turin, Grugliasco, Italy; [4]Department of Clinical and Biological Sciences, University of Turin, Orbassano, Italy; [5]Pediatric Oncology Branch, Oncogenomics Section, Center for Cancer Research, National Institutes of Health (NIH), Bethesda, United States; [6]Department of Molecular Biotechnology and Health Sciences, University of Turin, Turin, Italy; [7]Department of Experimental Oncology, Fondazione IRCCS Istituto Nazionale Tumori, Milan, Italy; [8]Department of Pathology, Boston Children's Hospital and Harvard Medical School, Boston, United States

*For correspondence: riccardo.taulli@unito.it

Present address: †Samantha Dickson Brain Cancer Unit, University College London Cancer Institute, London, United Kingdom; ‡Massachusetts General Hospital Cancer Center, Harvard Medical School, Charlestown, United States; §Department of Biological Sciences and the Center for Neuroscience Research, University at Albany, State University of New York, Albany, United States; ¶The Francis Crick Institute, Mill Hill Laboratory, London, United Kingdom; **School of Life Sciences, Ecole Polytechnique Fédérale de Lausanne (EPFL) & Swiss Institute of Bioinformatics, Lausanne, Switzerland

Competing interests: The authors declare that no competing interests exist.

**Abstract** Embryonal Rhabdomyosarcoma (ERMS) and Undifferentiated Pleomorphic Sarcoma (UPS) are distinct sarcoma subtypes. Here we investigate the relevance of the satellite cell (SC) niche in sarcoma development by using Hepatocyte Growth Factor (HGF) to perturb the niche microenvironment. In a *Pax7* wild type background, HGF stimulation mainly causes ERMS that originate from satellite cells following a process of multistep progression. Conversely, in a *Pax7* null genotype ERMS incidence drops, while UPS becomes the most frequent subtype. Murine EfRMS display genetic heterogeneity similar to their human counterpart. Altogether, our data demonstrate that selective perturbation of the SC niche results in distinct sarcoma subtypes in a Pax7 lineage-dependent manner, and define a critical role for the Met axis in sarcoma initiation. Finally, our results provide a rationale for the use of combination therapy, tailored on specific amplifications and activated signaling pathways, to minimize resistance emerging from sarcomas heterogeneity.

## Introduction

Rhabdomyosarcoma (RMS), the most common soft tissue sarcoma of childhood, is a rare but aggressive malignancy (*Hawkins et al., 2013*; *Saab et al., 2011*). Rhabdomyoblasts are positive for markers of muscle stem cells (satellite cells) such as Pax7 (*Tiffin et al., 2003*), myoblasts (MyoD and Myogenin) and differentiated skeletal muscle (Desmin and Myosin Heavy Chain MHC) (*Merlino and Helman, 1999*; *Morotti et al., 2006*). Thus, the identification of the RMS cell of origin is still debated

**eLife digest** Soft tissue sarcomas are rare cancers that originate in tissues such as muscles, tendons, cartilage and fat. These cancers are further classified into subtypes based on their appearance. For example, rhabdomyosarcoma cells resemble the cells that normally develop into muscle, while other soft tissue tumors that do not look like a distinct cell type are called undifferentiated pleomorphic sarcomas. Recent experiments have suggested that although these subtypes appear different, they may both arise from the cells that build muscles. However, this had not been confirmed.

Morena et al. investigated whether changing the environment – also known as the "niche" – of muscle stem cells could influence what type of sarcoma developed in mice that were prone to cancer. Normally muscle stem cells in an adult only regenerate injured muscles, and need to receive the correct cues before they divide. Among these cues is a protein called Hepatocyte Growth Factor (or HGF for short), which is produced by cells in the muscle stem cells' niche.

Morena et al. engineered mice so that the production of HGF in the muscles could be switched on or off at will. Mice that were already prone to cancer and produced a lot of HGF tended to develop rhabdomyosarcomas. However, when HGF was turned on in similar mice that also lacked normal muscle stem cells, the resulting sarcomas were predominantly undifferentiated pleomorphic sarcomas. These data indicate that rhabdomyosarcomas probably originate from muscle stem cells, whereas undifferentiated pleomorphic sarcomas develop from other cells in the niche.

Lastly, Morena et al. studied the sarcomas in their mice in more detail and observed that, similar to what has been found in human rhabdomyosarcomas, individual tumors had different genetic mutations. These differences make it difficult to treat sarcomas with a single anti-cancer drug. However, the new results suggest that a combination of targeted drugs may prove effective in blocking tumor growth and in preventing resistance.

and remains elusive. On the other hand undifferentiated pleomorphic sarcoma (UPS), which is one of the most common subtypes of adult soft tissue sarcoma, is characterized by a lack of tissue-specific differentiation markers. UPS may originate from cells of different lineages that converge into a common undifferentiated histological presentation, or may derive by a process of de-differentiation from more committed sarcomas (*Weiss and John, 2007*).

Histopathological classification of RMS includes two major subgroups, the alveolar (ARMS) and the embryonal (ERMS) subtype. ARMS are 20% of all newly diagnosed RMS. Although rarer, they are often metastatic at diagnosis and have a poor prognosis. 80% of ARMS are characterized by the *PAX3/7-FOXO1* chromosomal translocation (*Mercado and Barr, 2007*). The remaining ones, negative for the translocation, are indistinguishable from ERMS at the molecular level (*Williamson et al., 2010*).

In the last decades standard treatments including radiotherapy, chemotherapy and surgery, have not significantly improved RMS and UPS patient survival. Thus, novel effective precision-based therapeutic approaches are required. While the mutational status of UPS has been only sporadically analyzed (*Li et al., 2015*), the comprehensive genomic and epigenetic landscape of RMS tumors was recently described (*Chen et al., 2013*; *Seki et al., 2015*; *Shern et al., 2014*). These studies highlight the difference between ARMS and ERMS in terms of mutational load. While ARMS carry only a few genetic lesions in addition to the pathognomonic ones, the ERMS subtype is highly heterogeneous, with recurrent mutations/copy number variations in genes coding for tyrosine kinase receptors (RTKs) and their downstream effectors (RAS and PIK3CA).

The early onset of ERMS, concomitant with a period of intense muscle growth and their positivity for Pax7, suggest that the muscle stem cells (satellite cells, SCs) could be at the origin of this subtype. Quiescent SCs express the Met receptor (*Allen et al., 1995*) and are found adherent to the muscle fibers in a specialized sublaminar microenvironment called SCs niche. The niche microenvironment controls their fate by orchestrating the homeostatic balance between stem cell quiescence and activation. Upon injury, the niche releases HGF, which is one of the extrinsic signals involved in SC activation and proliferation (*Allen et al., 1995*; *Tatsumi et al., 1998*; *Thomas et al., 2015*).

Although others have linked injury with sarcoma also through activation of Met signaling (*Sharp et al., 2002*; *Tremblay et al., 2014*; *Van Mater et al., 2015*), we here describe a unique model aimed at investigating the effect of HGF-mediated SC niche perturbation in sarcoma development and maintenance. Specifically, HGF expression was confined to the SC niche and could be temporally regulated by Doxycycline (Dox). In a wild type background, HGF production promoted only limited activation of satellite cells, without inducing an overt phenotype. Conversely, in a *Cdkn2a* null background all mice developed sarcoma, 92% of which classified as ERMS and only 8% as UPS. Genetic ablation of the muscle stem cells (obtained by moving the system in a *Pax7* null background) strongly influenced the sarcoma subtype. In this different genetic setting the majority of tumors were classified as UPS, suggesting that in the absence of satellite cells, fibroblasts resident in the SC niche were the more susceptible population to HGF-mediated perturbation. Finally, we investigated the relevance of novel therapeutic approaches using our preclinical model of sarcoma. The vast majority of tumors grew in a HGF/Met-independent manner and were genetically heterogeneous. Tumor cells were sensitive to Met inhibitors only in the rare cases harboring *Met* amplification, but the continuous treatment with a single agent resulted in selection and expansion of resistant clones. However, the use of a combination of drugs hitting different targets was effective in bypassing resistance. Altogether, our data show that perturbation of the SC niche with HGF can promote distinct sarcoma subtypes in a Pax7 lineage-dependent manner, thus offering a possible explanation of why ERMS and UPS are part of a tumor continuum. Finally, the use of our model for the preclinical assessment of targeted therapy revealed that combination, rather than single agent treatment, could be more effective in treating genetically heterogeneous sarcomas.

## Results

### Met and satellite signatures are both preferentially associated with the ERMS subtype, while UPS show high Met and fibroblast scores

At variance with other sarcoma subtypes, a satellite cell-like signature is considered a hallmark of ERMS (*Hatley et al., 2012*; *Rubin et al., 2011*) while the complete absence of tissue-specific markers in UPS suggests an origin from early mesenchymal precursors. By taking advantage of previously validated Met and satellite signatures (*Fukada et al., 2007*; *Bertotti et al., 2009*; *Pallafacchina et al., 2010*) we performed unsupervised clustering analysis of two large panels of human primary RMS (*Davicioni et al., 2009*; *Williamson et al., 2010*). We selected 5 clusters subdivisions to segregate normal muscle apart from RMS samples. Interestingly, in both datasets the Met signature was more effective than the satellite signature in distinguishing ERMS from ARMS (*Figure 1A*). We determined the Met score, which is essentially the sum contribution of the expression values of the signature genes and is a measure of the level of Met activity in each sample. The Met score was higher in RMS compared to muscle (*Figure 1B*) and in ERMS rather than ARMS (*Figure 1C*). The satellite score, measured in an analogous way, was also able to discriminate RMS from muscle, but did not consistently distinguish ERMS from ARMS (*Figure 1—figure supplement 1A,B*). We then analyzed the top 25% of patients with high Met and satellite scores. Interestingly, only by using the Met score the first top quartile was enriched in ERMS samples (*Figure 1D*). However, in the top quartiles there was a significant overlap of ERMS samples with both high Met and satellite scores (*Figure 1E*). Finally, we used the Met signature and a fibroblast signature (*ad hoc* calculated) to analyze a large human dataset covering different sarcoma subtypes (*Gibault et al., 2011*). Interestingly, the Met score was significantly enriched in UPS compared to other sarcomas, while the fibroblast signature distinguished UPS from RMS (*Figure 1F,G*).

Overall our bioinformatic analyses provide a basis to investigate the functional relationship between the HGF/Met axis and the SC niche in ERMS/UPS initiation and maintenance.

### A spatially and temporally restricted model of sarcoma based on SC niche perturbation

To generate a model of ERMS based on SC niche perturbation (herein defined MH) we conditionally modulated transgenic *Hgf* and eGFP expression using the Muscle Creatine Kinase-driven tTA (*Ckm*-Tet-Off) promoter (*Figure 2A*). MH mice, born at the expected Mendelian frequency, showed normal postnatal muscle growth (*Figure 2—figure supplement 1A,B*). Expression of *Hgf* and eGFP was

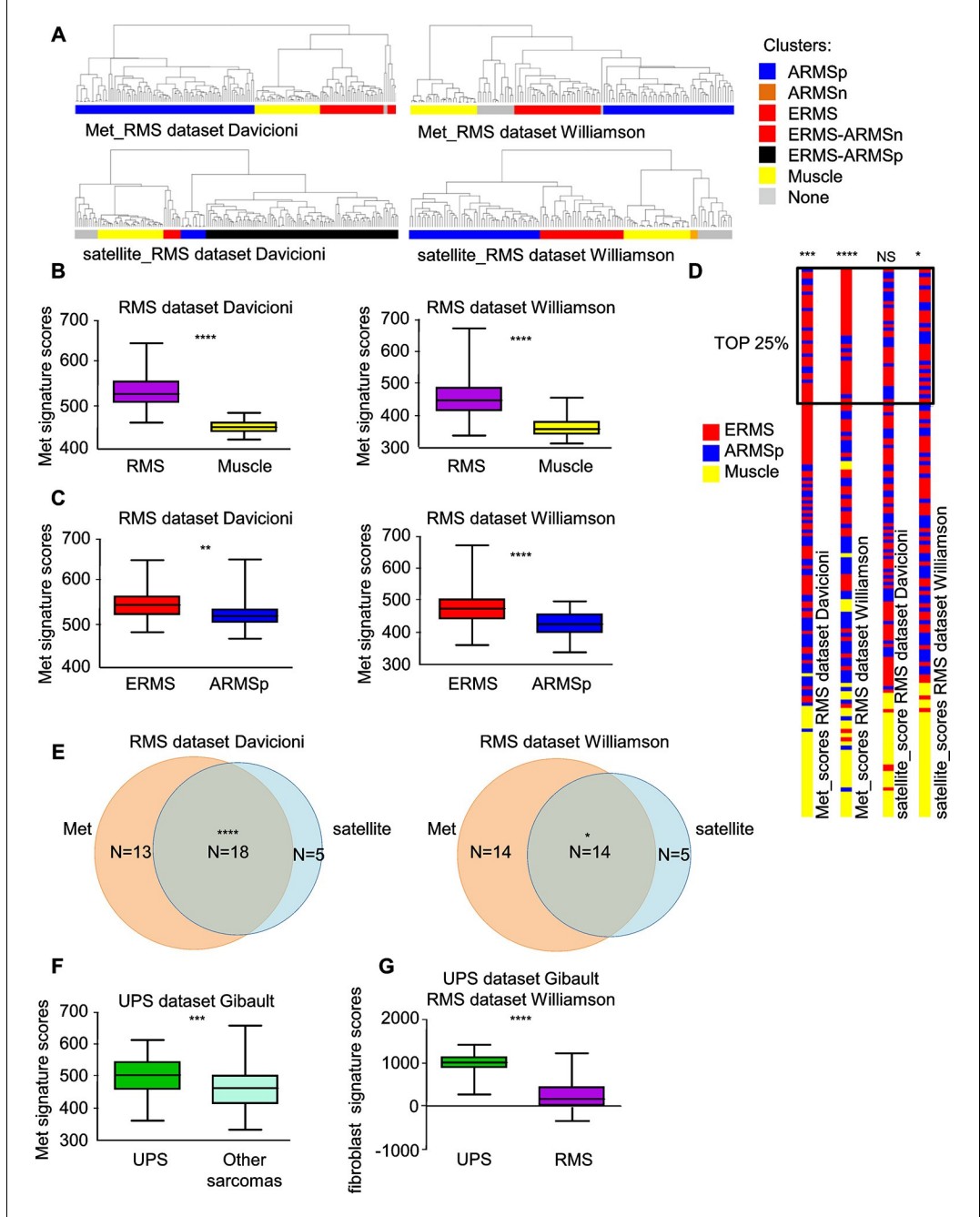

**Figure 1.** Met and satellite signatures are both preferentially associated with the ERMS subtype, while UPS show high Met and fibroblast scores. (A) Unsupervised hierarchical clustering of RMS samples according to the Met or satellite signature genes; colors highlight RMS subtype enrichment in single clusters (ARMSp: translocation positive; ARMSn: translocation negative). (B) Box-plot of the Met signature scores for RMS and muscles in the indicated datasets. ****p<0.0001 (t test). (C) Box-plot of the Met signature scores for ERMS and ARMSp in the indicated datasets. **p<0.01; ****p<0.0001 (t test). (D) Ranked distribution of RMS subtypes according to the indicated signature scores. The boxed area includes the top 25% samples. NSp>0.05; *p<0.05; ***p<0.001; ****p<0.0001 (hypergeometric test). (E) Venn diagrams of ERMS included in the box in D, showing a significant intersection between high Met and high satellite signatures. *p<0.05; ****p<0.0001 (hypergeometric test). (F) Box-plot of the Met signature scores for UPS and other sarcomas in the indicated dataset. ***p<0.001 (t test). (G) Box-plot of the fibroblast signature scores for UPS and RMS in the indicated datasets. ****p<0.0001 (t test).

The following figure supplement is available for figure 1:

**Figure supplement 1.** Satellite signature analysis in human RMS datasets.

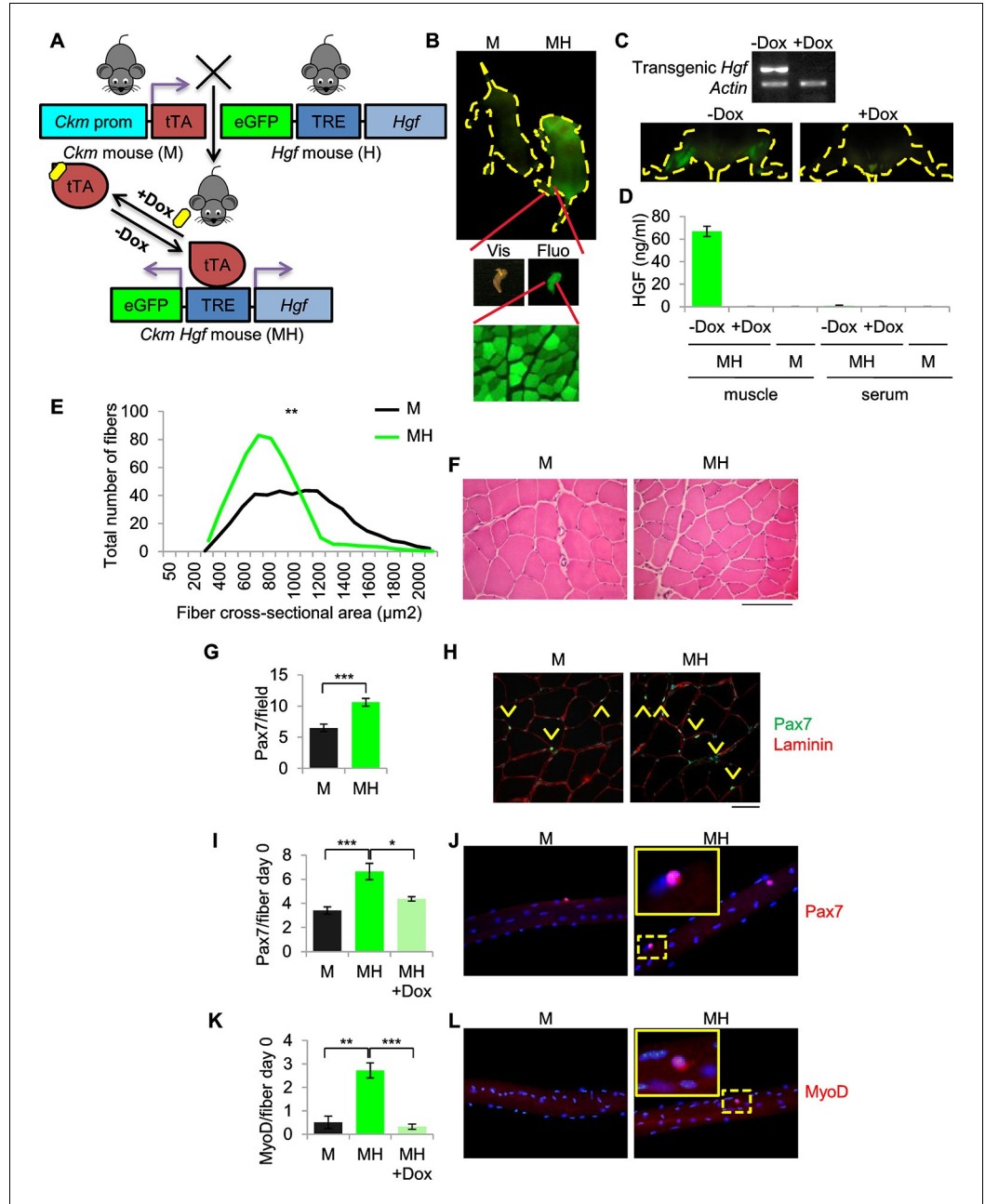

**Figure 2.** SC niche perturbation results in satellite cells activation. (**A**) Schematic representation of the system used to generate *Ckm*-Tet-Off *Hgf* (MH) mice. In the absence of Doxycycline (-Dox), tTA binds to the Tetracycline Responsive Element (TRE), inducing the expression of HGF and eGFP in skeletal muscle. (**B**) Upper panel: fluorescent image of P10 mice. Middle panel: MH hindlimb under visible (Vis) and fluorescent (Fluo) light. Lower panel: fluorescent image of a muscle cross-section of a MH mouse. (**C**) Upper panel: semi-quantitative RT PCR of the indicated genes on *Tibialis anterior* muscles. Lower panel: fluorescent images of hindlimbs from a MH mouse with or without Dox treatment. (**D**) HGF-ELISA quantification in muscle and serum extracts from Dox-treated or control mice. (**E**) Distribution of *Tibialis anterior* cross-sectional areas showing a leftward shift in MH mice relative to their respective M controls. The mean value for MH mice was $585 \pm 37 \ \mu m^2$ (n=8); the mean value for control M mice was $844 \pm 63 \ \mu m^2$ (n=10). **p<0.01 (*t* test). (**F**) Representative H&E staining of *Tibialis anterior* cross sections. (Scale bar = 100 μm). (**G**) Quantification (mean ± SEM) of Pax7 positive cells/field in *Tibialis anterior* sections (M mice n=7; MH mice n=6). ***p<0.001 (*t* test). (**H**) Representative immunofluorescence of Pax7 (green), Laminin (red) and DAPI (blue) staining in G. Arrowheads indicate Pax7-positive cells. (Scale bar = 50 μm). (**I**) Quantification (mean ± SEM) of Pax7 positive cells/fiber after single fiber isolation (M mice n=13; MH mice n=12; MH +Dox n=2). *p<0.05; ***p<0.001 (*t* test). (**J**) Representative immunofluorescence of Pax7 (red) and DAPI (blue) staining in I. (**K**)
*Figure 2 continued on next page*

*Figure 2 continued*

Quantification (mean ± SEM) of MyoD-positive cells/fiber after single fiber isolation (M mice n=3; MH mice n=3; MH +Dox n=4). **p<0.01; ***p<0.001 (*t* test). (L) Representative immunofluorescence of MyoD (red) and DAPI (blue) staining in K. Dashed areas are shown at threefold magnification.

The following figure supplement is available for figure 2:

**Figure supplement 1.** SC niche perturbation results in satellite cells activation.

restricted to skeletal muscle and was Dox-dependent (*Figure 2B,C*). While HGF was measurable in transgenic muscle, it was undetectable in blood (*Figure 2D*). Muscle sections showed a modest reduction of fiber size (*Figure 2E,F*, *Figure 2—figure supplement 1C*). However, Pax7 immunofluorescence revealed a higher number of satellite cells compared to control littermates (*Figure 2G,H*). More Pax7-positive cells were also detected on freshly isolated single myofibers (*Figure 2I,J*). Surprisingly, at time 0 some satellite cells were MyoD-positive, indicating basal activation (*Figure 2K,L*). These differences were Dox-dependent (*Figure 2I,K*) and lasted for three days in culture (*Figure 2—figure supplement 1D,E*). Furthermore, myofibers isolated from MH mice contained more myonuclei than controls (*Figure 2—figure supplement 1F*). MyoD-positive cells were also detectable in transgenic muscle sections (*Figure 2—figure supplement 1G*), indicating the presence of activated satellite cells also in vivo.

## Loss of homeostatic balance between SC proliferation and differentiation results in a multistep model of ERMS development

Homozygous deletions of the *CDKN2A* locus has been reported in the majority of human tumors including sarcomas (*Miller et al., 1997*; *Roussel, 1999*; *Ruas and Peters, 1998*). Specifically, CGH analysis revealed that more than 17% of UPS exhibited the loss of the locus, while up to 39% of the UPS did not express the corresponding p14 protein (*Pérot et al., 2010*). The frequent inactivation of the locus has been confirmed by an additional study (*Simons et al., 2000*) reporting the deletion in 32% of UPS cases. Furthermore, the genetic inactivation of the *CDKN2A* locus in RMS has been extensively reported (*Chen et al., 2007*; *Obana et al., 2003*; *Paulson et al., 2011*; *Seki et al., 2015*). Thus we moved our model into a *Cdkn2a* (alias *Ink4a/Arf*) null genetic background (herein defined MHI-null, *Figure 3—figure supplement 1A*). Approximately 30% of all MHI-null mice displayed an apparently hypertrophic phenotype (*Figure 3A–C*). All of these mice quickly developed tumors and the histology revealed a picture of multistage progression (*Figure 3D*, stage 1 to 4). In the periphery of the tumor (*Figure 3D*, stage 1) proliferating (Ki67-positive) satellite cells, surrounding old fibers, were mixed with more mature MyoD- and Myogenin-positive cells. These cells were intermingled with areas of neomyogenesis, including small fibers with central nuclei, positive for embryonal MHC (eMHC). Immunofluorescence confirmed the presence of Desmin-positive centronucleated fibers contiguous to old fibers (*Figure 3E*, left panel). At this early stage, Pax7-positive cells were detectable beneath the basal lamina of the newly formed fibers, suggesting that satellite cells retained the ability to return to quiescence (*Figure 3E*, right panel). Closer to the tumor mass the expanded population of proliferating satellite cells appeared to take over, while the number of newly formed fibers was reduced (*Figure 3D*, stages 2 and 3). The central area of the tumor consisted of tissue with high cellularity and no further evidence of terminal myogenic differentiation (*Figure 3D*, stage 4). Accordingly, the tumor bulk was eGFP-negative indicating that the cells were undifferentiated (*Figure 3B*, *Figure 3—figure supplement 1B*). Morphologically, small round cells were associated with elongated polygonal cells or with cells displaying pleomorphic nuclei (*Figure 3D*, stage 4). The histological markers (Pax7, MyoD and Myogenin, *Figure 3D*, stage 4) and the anatomical localization (trunk, neck and occasionally the orbit) (*Figure 3—figure supplement 1B*), were compatible with the ERMS subtype. MHI-null mice developed tumors with a short latency (3.95 months), while mice heterozygous for the *Cdkn2a* locus (MHI-het) showed delayed tumor formation (6 months) (*Figure 3—figure supplement 1C*). PCR analysis revealed that the majority of the tumors in MHI-het mice had lost the wild type allele (*Figure 3—figure supplement 1D*). To exclude the involvement of fetal myoblasts in ERMS development, MHI-null mice were maintained under Dox treatment until P10. Given the persistence of Dox in the tissues, transgenic *Hgf* and eGFP

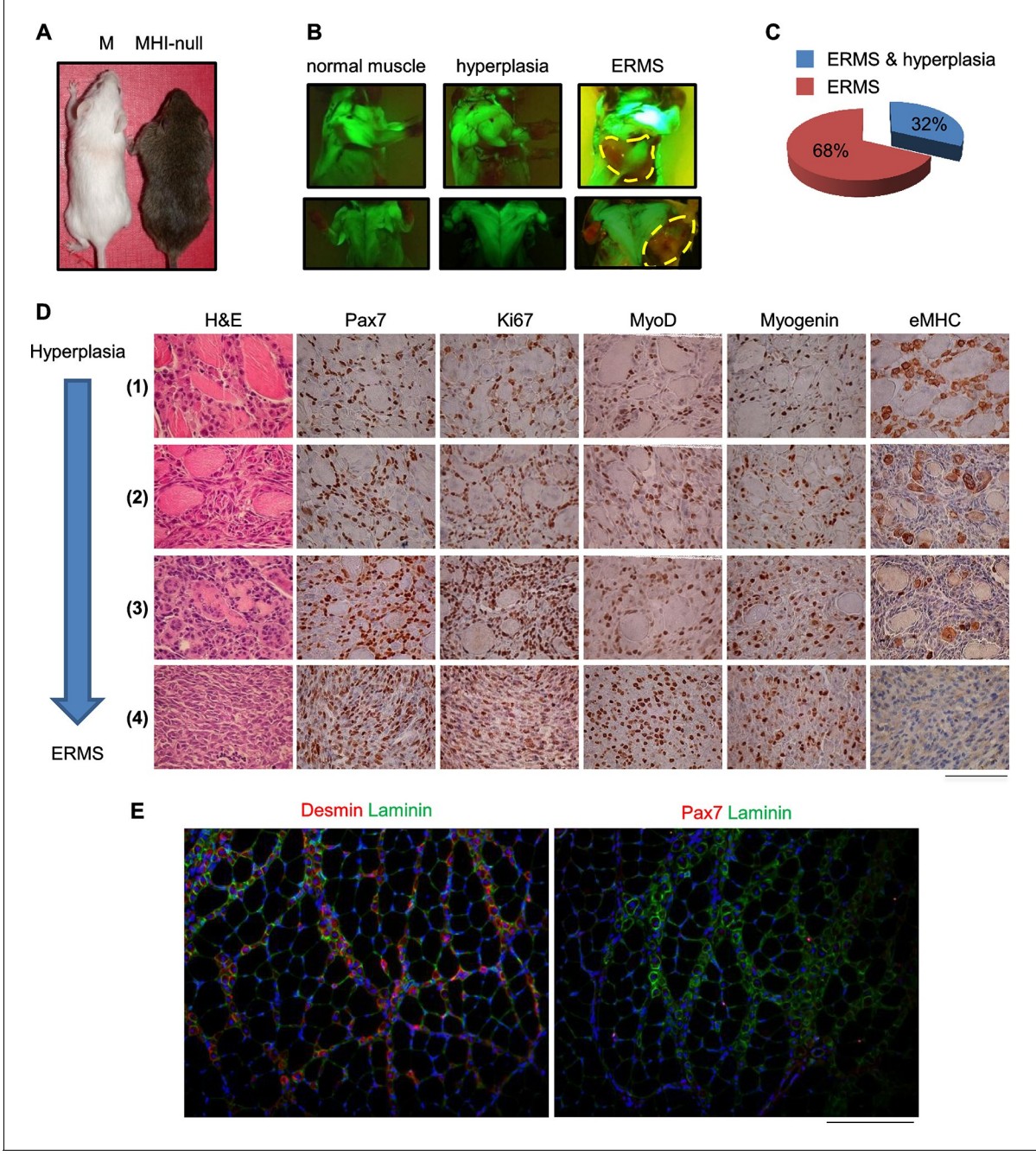

**Figure 3.** SC niche perturbation in *Cdkn2a*-null mice results in a multi-step model of ERMS development. (**A**) Representative image of a control (M) and a hyperplastic mouse (MHI-null). (**B**) Representative fluorescent images of MHI-null trunk muscles at different stages of progression: normal muscle, hyperplasia and ERMS. (**C**) Distribution of MHI-null mice developing ERMS with or without hyperplasia. (**D**) H&E staining and representative immunohistochemical analysis of MHI-null specimens collected at different stages of tumor progression (from hyperplasia to ERMS, stage 1–4). (Scale bar = 100 μm). (**E**) Representative immunofluorescence on MHI-null hyperplastic muscle sections. Pax7 (red), Laminin (green), Desmin (red) and DAPI (blue). (Scale bar = 100 μm).

The following figure supplements are available for figure 3:

**Figure supplement 1.** SC niche perturbation in *Cdkn2a*-null mice results in sarcoma development.

**Figure supplement 2.** Evaluation of *Hgf* expression and mortality in MHI-null mice treated with Dox from conception to P10.

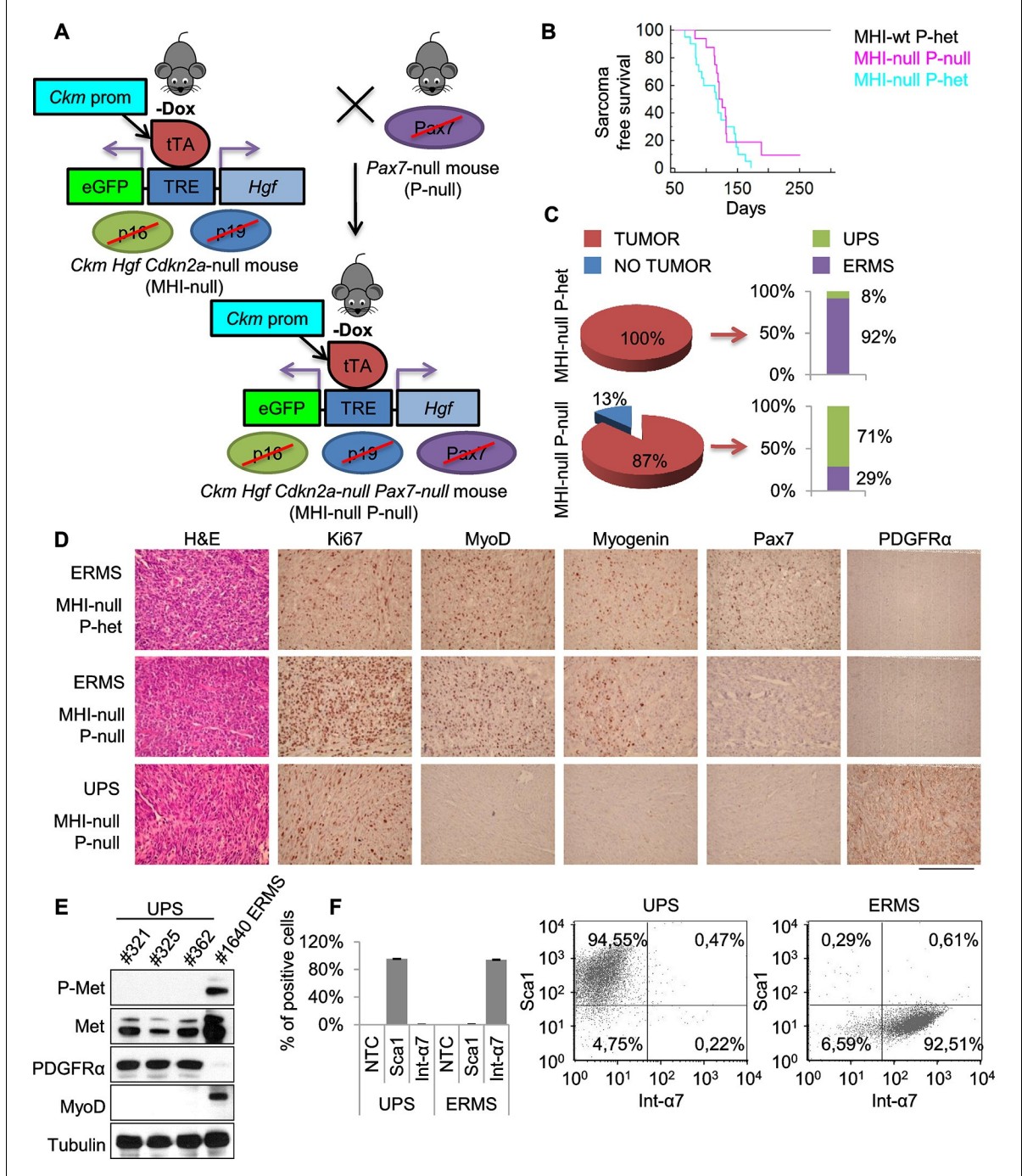

**Figure 4.** Genetic ablation of *Pax7* results mainly in UPS development. (**A**) Schematic representation of the system used to generate *Ckm Hgf Cdkn2a*-null *Pax7*-null (MHI-null P-null) mice. (**B**) Kaplan-Meier curve showing the relative sarcoma-free survival in the indicated genotypes. (**C**) Penetrance (pie charts) and histological distribution (bar graphs) of tumors. (**D**) Representative immunohistochemical analysis of MHI-null tumors in *Pax7* heterozygous or null genetic background. UPS: undifferentiated pleomorphic sarcoma. (Scale bar = 250 µm). (**E**) Western blot analysis of murine UPS and ERMS probed for the indicate proteins. (**F**) Sca1 and Int-α7 expression analysis in primary murine UPS and ERMS (left panel, single staining; center and right panels, double staining).

expression became detectable from day 40 onward (*Figure 3—figure supplement 2A,B*), when only resident muscle stem cells are present. In this cohort of mice, tumor development occurred with the previously described latency and incidence (*Figure 3—figure supplement 2C*).

## SC niche perturbation in a *Pax7*-deficient background mainly results in UPS development

It has been shown that ERMS and UPS are part of a tumor continuum in terms of histological presentation and expression profiling (*Rubin et al., 2011*). Intriguingly, the SC niche, in addition to muscle stem cells, also contains resident muscle fibroblasts that are responsible for extracellular matrix production within skeletal muscle fibers (*Thomas et al., 2015*). To assess the contribution of these two cell types to sarcomagenesis we moved the MHI-null mice in a *Pax7* null genetic background. *Pax7*-deficient C57/B6 mice are viable at birth, but progressively die in 2–3 weeks (*Mansouri et al., 1996*; *Seale et al., 2000*). In these mice the number of satellite cells is reduced at birth, and further declines during postnatal development. However, in a mixed genetic background, between 5 and 10% of mutant animals survive into adulthood (*Oustanina et al., 2004*). Thereby, we generated a cohort of MHI-null *Pax7*-null mice (herein referred as MHI-null P-null) (*Figure 4A*) sufficient to determine the effect of the *Pax7* null background on sarcoma development. The mixed background had no effect on sarcoma incidence and latency (*Figure 4B*). However MHI-null P-null mice showed a drastic reduction in ERMS incidence (*Figure 4C*). The MHI-null P-null tumors were characterized by cells displaying a variegate morphology, that occasionally included the presence of visible cross-striations. Notably, immunohistochemistry revealed the positivity for MyoD and Myogenin in less than 30% of cases, which were classified as ERMS (*Figure 4C,D*). Interestingly, the majority of tumors (more than 70%) presented extensive cellular heterogeneity from round to spindle-shape cells, with nuclear appearance ranging from hyperchromatic to pleomorphic. The complete absence of tissue-specific markers of differentiation categorized these tumors as UPS (*Figure 4C,D*). The switch in sarcoma subtype is in line with the expansion of myofibroblasts observed in *Pax7*-null muscles upon exposure to a regenerative microenvironment (*Maltzahn et al., 2013*). Accordingly, while both UPS and ERMS tumors expressed Met, only UPS were positive for the pan-fibroblast marker PDGFRα (*Figure 4D,E*), and FACS analysis on freshly isolated tumors revealed that they were Sca1-positive and Integrin-α7-negative (*Figure 4F*), indicating a mesenchymal-like phenotype (*Driskell et al., 2013*; *Guarnerio et al., 2015*; *Joe et al., 2010*). In contrast, the ERMS were Sca1-negative and Integrin-α7-positive, confirming the myogenic identity of this subtype. Altogether our data clearly indicate that perturbation of the SC niche microenvironment can give rise to distinct sarcoma subtypes in a Pax7 lineage-dependent manner, and provide evidence for a fibroblast cell origin of UPS.

## SC niche perturbation results in heterogeneous tumors, with only a subset displaying sensitivity to Met or PI3K pathway inhibition

Genetically engineered animal models represent a robust preclinical platform to investigate the efficacy of novel treatments, among which targeted therapies and combination therapies are the most attractive (*Day et al., 2015*). In our model transgenic *Hgf* comes from differentiated muscle fibers. Thus the tumor mass is no longer exposed to transgenic HGF for lack of differentiated muscle cells, as shown by the absence of expression of the eGFP reporter in the tumors (see *Figure 3B*, *Figure 3—figure supplement 1B*). Accordingly, Dox-mediated *Hgf* downmodulation did not impair tumor growth (*Figure 5—figure supplement 1A,B*) indicating that transgenic HGF production was not essential for tumor maintenance. Although all primary murine tumors retained basal Met expression, in a subset of ERMS (which later turned out to be *Met*-amplified), Met was present at much higher level and overexpression was accompanied by receptor phosphorylation (*Figure 5A,B*, and *Figure 5—figure supplement 2B*). On the other hand, at earlier stages, paracrine HGF stimulation promoted receptor downregulation (*Figure 5—figure supplement 2A*) and this resulted in a difficult detection of Met and phospho-Met (*Figure 5—figure supplement 2B*).

Despite extensive attempts to stabilize cell lines from both sarcoma subtypes, we obtained a significant number of cell lines only from ERMS. To identify signaling pathways involved in tumor maintenance, we tested a panel of 18 ERMS lines with receptor tyrosine kinase (RTK) phosphoarrays. Met activation was detected in three cell lines (#187R, #1640 and #1796, 17% of the total) (*Figure 5—figure supplement 3A,B*) that all turned out to be sensitive to Met inhibition (see below). Notably, Met had already been found overexpressed and phosphorylated in the corresponding primary tumor specimens (*Figure 5A,B* and not shown for tumor #187R) ruling out the possibility that Met activation resulted from in vitro culture conditions. Comparative genomic hybridization (CGH) revealed amplification of the *Met* locus in three samples (#1640, #1671 and #187R), and gain of the entire

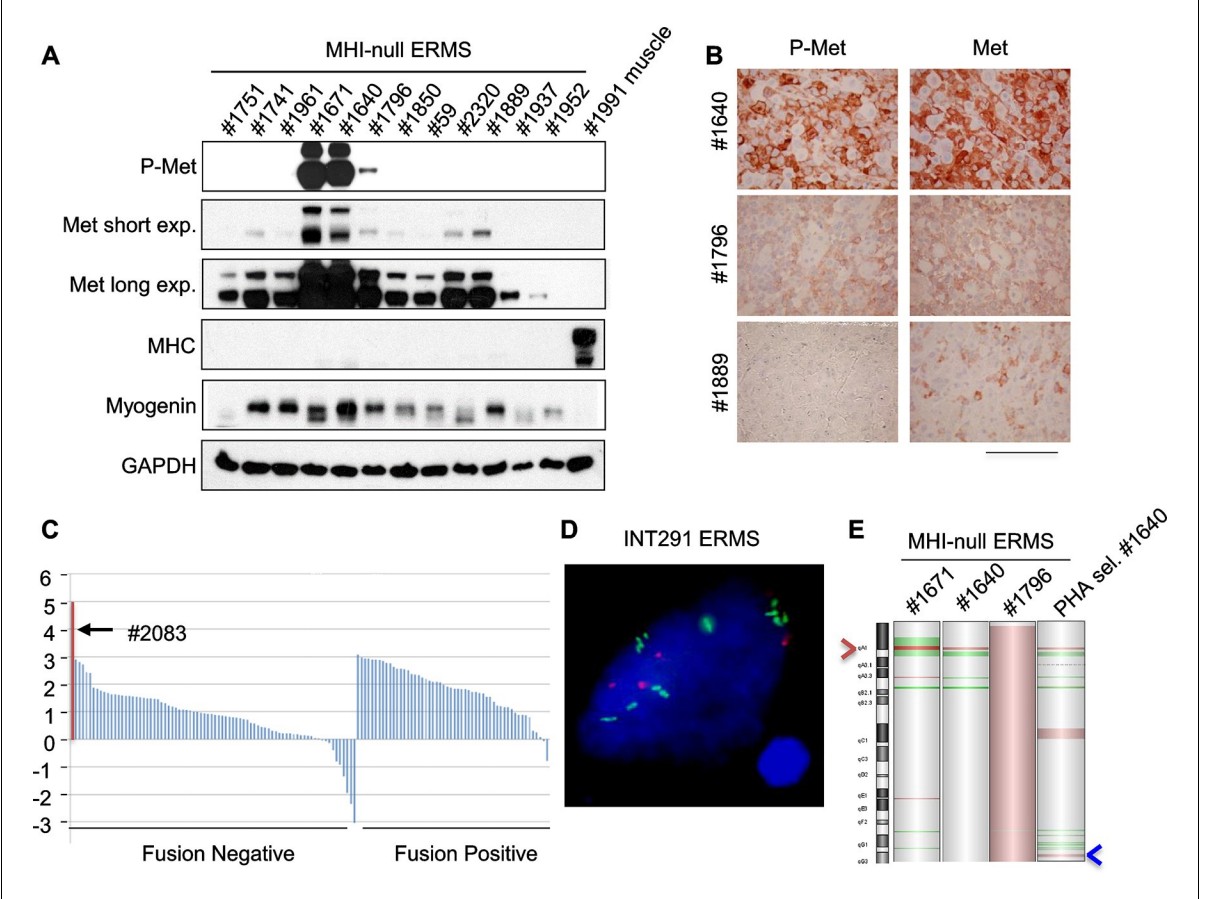

**Figure 5.** Only a minor fraction of ERMS harbors *MET* amplification. (**A**) Western blot analysis in a panel of primary murine MHI-null ERMS probed for the indicate proteins. (**B**) Immunohistochemical analysis of primary murine MHI-null ERMS shown in A, probed for Met and phosphorylated Met. (Scale bar = 100 μm). (**C**) Waterfall plot showing the *MET* Z-score expression from a panel of previously characterized human RMS tumors (*Shern et al., 2014*) (http://pob.abcc.ncifcrf.gov/cgi-bin/JK). The exceptionally high *MET* expression in Patient 2083 (highlighted) is associated with focal amplification of chromosome 7q31.2 (*Shern et al., 2014*). (**D**) FISH for *MET* in a human ERMS sample (INT291) characterized by small cluster signals (with an average of 9 signals per cell; *MET* is labeled in green and centromeres in red). (**E**) CGH analysis of murine MHI-null ERMS. Red indicates copy number gain, green indicates copy number loss. Arrowheads mark the *Met* (red) and *Kras* (blue) loci.

The following figure supplements are available for figure 5:

**Figure supplement 1.** Transgenic *Hgf* downmodulation does not impair tumor growth.

**Figure supplement 2.** Determination of the level of Met protein and phosphorylation in rhabdomyosarcomagenesis.

**Figure supplement 3.** Only a minor fraction of murine ERMS displays Met activation and amplification.

**Figure supplement 4.** Evaluation of Met expression in human RMS datasets.

chromosome 6 in another one (#1796) (*Figure 5E*, *Figure 7H*). Amplification of the *Met* locus was confirmed by real-time PCR on genomic DNA (*Figure 5—figure supplement 3C*, *Figure 7—figure supplement 1D*).

Analysis of the level of *MET* transcript in human RMS datasets (*Davicioni et al., 2009*; *Williamson et al., 2010*) confirmed the variability in terms of Met expression (*Figure 5—figure supplement 4A,B*). Similar results were obtained by the analysis of RNA sequencing data on a different large cohort of human patients (*Figure 5C*). The only ERMS sample where Met expression rose above all others (#2083) was found to harbor *MET* amplification (*Shern et al., 2014*). Interestingly,

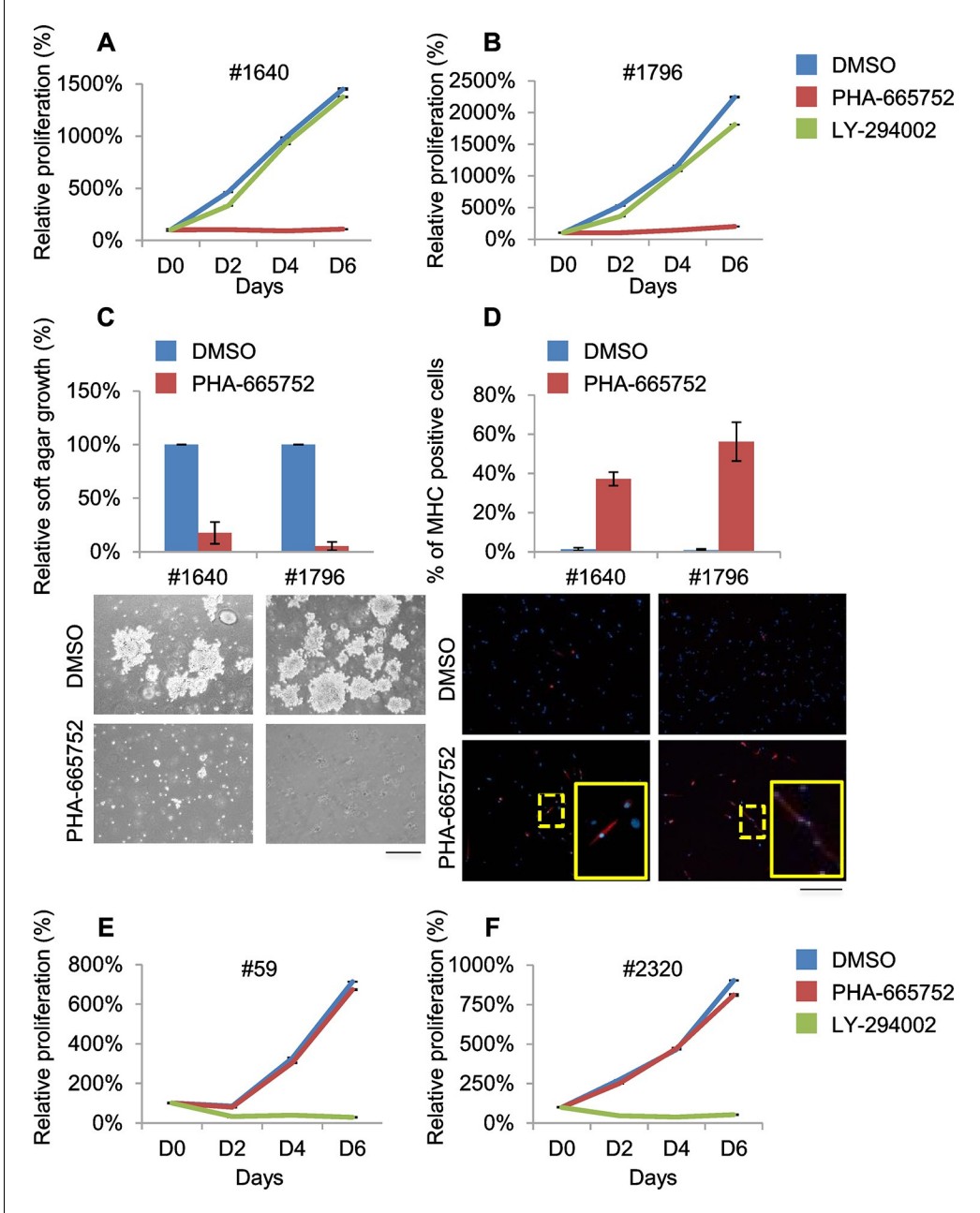

**Figure 6.** Murine ERMS are genetically heterogeneous, with subsets specifically sensitive to pharmacological inhibition of distinct drivers. (A, B) Proliferation analysis (mean ± SD) of the indicated MHI-null ERMS cells treated with Met (PHA) and PI3K (LY) inhibitors. The number of cells at day 0 was set at 100%, representative assay of at least 2 independent experiments. (C) Quantification and representative images of soft agar growth assays of cells treated with PHA. The number of colonies obtained from cells maintained in DMSO control was set at 100% (3 independent experiments, mean ± SEM). (Scale bar = 500 µm). (D) MHC expression analysis and representative MHC immunostaining of cells treated with PHA for 3 days (3 independent experiments, mean ± SEM). Dashed areas are shown at threefold magnification. (Scale bar = 250 µm). (E, F) Proliferation analysis (mean ± SD) of the indicated MHI-null ERMS cells treated with Met (PHA) and PI3K (LY) inhibitors. The number of cells at day 0 was set at 100%, representative assay of at least 2 independent experiments.

The following figure supplement is available for figure 6:

**Figure supplement 1.** Murine ERMS are genetically heterogeneous, with subsets specifically sensitive to pharmacological inhibition of distinct drivers.

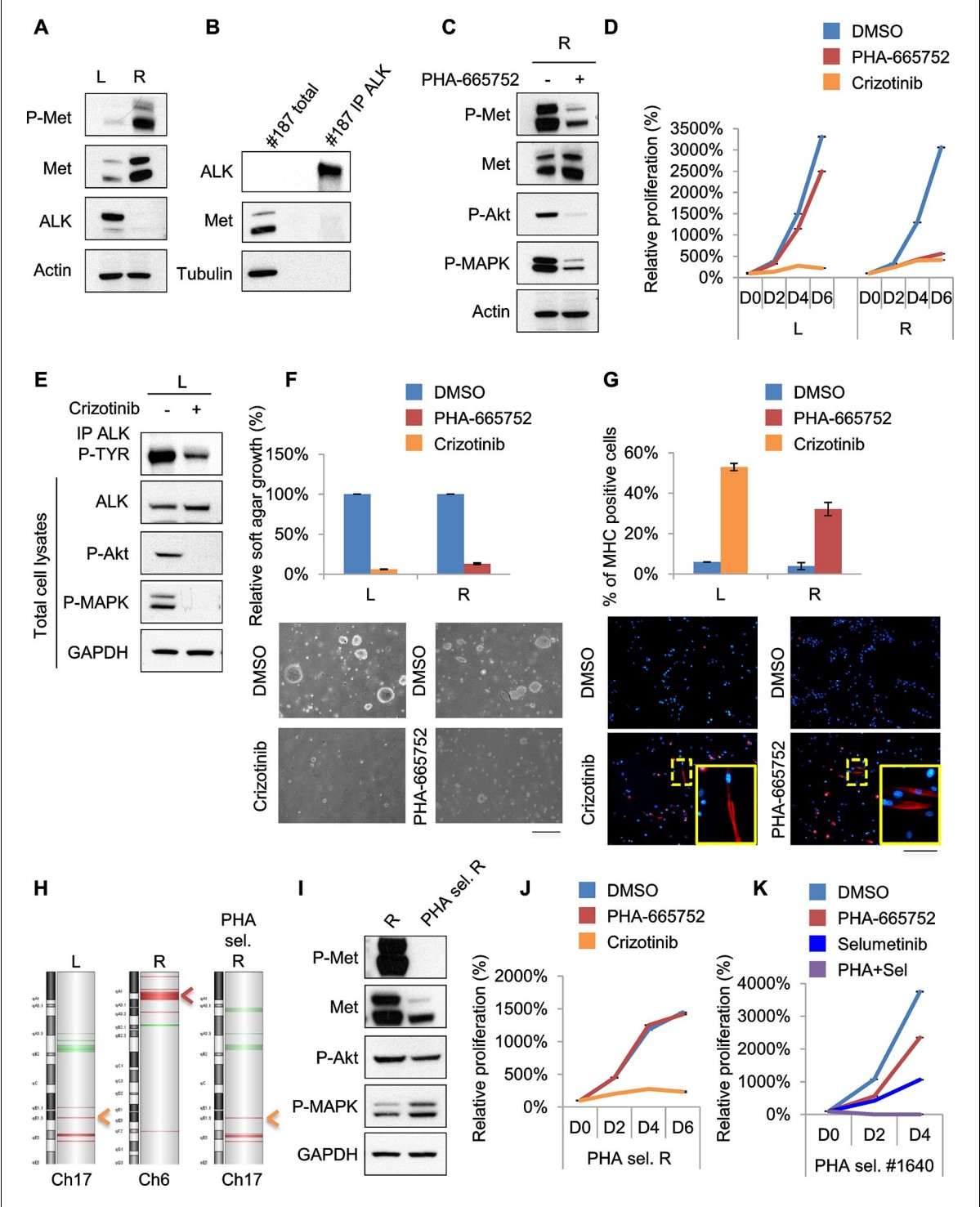

**Figure 7.** Treatments based on combination therapy can be effective in preventing ERMS recurrence and clonal evolution. (**A**) Western blot analysis on the indicated proteins in L and R cells. (**B**) Western blot on total lysate and ALK-immunoprecipitated fraction of #187 ERMS tumor. (**C**) Western blot analysis of R cells treated with PHA for 2 hr. (**D**) Proliferation analysis (mean ± SD) of L and R cells treated with the indicated inhibitors. The number of cells at day 0 was set at 100%, representative assay of at least 2 independent experiments. (**E**) Western blot analysis on total cell lysate and ALK-immunoprecipitated fraction of L cells, treated with 500 nM Crizotinib for 2 hr. (**F**) Quantification and representative images of soft agar growth assays of L and R cells treated with the indicated inhibitors. The number of colonies obtained from cells in DMSO control was set at 100% (2 independent experiments, mean ± SEM). (Scale bar = 500 μm). (**G**) MHC expression analysis and representative MHC immunostaining of L and R cells, treated with the indicated inhibitors for 3 days (2 independent experiments, mean ± SEM). Dashed areas are shown at threefold magnification. (Scale bar = 250 μm). *Figure 7 continued on next page*

*Figure 7 continued*

(**H**) CGH analysis of MHI-null ERMS cell lines. Red indicates copy number gain, green indicates copy number loss. Arrowheads mark *Alk* (orange) and *Met* (red) loci. (**I**) Western blot analysis of R and PHA-selected R cells. (**J**) Proliferation analysis (mean ± SD) of PHA-selected R cells treated with the indicated inhibitors. The number of cells at day 0 was set at 100%, representative assay of at least 2 independent experiments. (**K**) Proliferation analysis (mean ± SD) of PHA-selected #1640 cells treated with the indicated inhibitors. The number of cells at day 0 was set at 100%, representative assay of at least 2 independent experiments.

The following figure supplement is available for figure 7:

**Figure supplement 1.** Treatments based on combination therapy can be effective in preventing ERMS recurrence and clonal evolution.

another example of human ERMS carrying *MET* amplification emerged from an independent FISH analysis of a small set (8 cases) of ERMS (*Figure 5D*). Overall murine and human data indicate a considerable variability of Met expression and rare *Met* amplification.

Treatment of #1640 and #1796 cell lines (#1671 cells were unfortunately lost) with the Met-specific inhibitor PHA-665752 (herein referred as PHA) resulted in the abrogation of Met signaling (*Figure 6—figure supplement 1A*), in the arrest of cell proliferation (*Figure 6A,B*), in the accumulation of cells in the G0/G1 phase of the cell cycle (*Figure 6—figure supplement 1B*), in the impairment of anchorage-independent growth (*Figure 6C*) and in the induction of apoptosis (*Figure 6—figure supplement 1C*). Interestingly, the rare surviving cells were fully differentiated (*Figure 6D*). Finally, nude mice bearing ERMS tumors derived from both cell lines were treated with Crizotinib, a well-tolerated dual Met and ALK inhibitor, approved for the treatment of patients with metastatic *ALK*-positive NSCLC (*Rodig and Shapiro, 2010*). Crizotinib administration to mice with palpable tumors blocked ERMS growth and reduced the tumor volume, without symptoms of toxicity (*Figure 6—figure supplement 1D*).

RTK downstream effectors are often dysregulated in ERMS. Accordingly, some ERMS cases of mutated phosphatidylinositol 3 kinase A (PIK3CA) have been reported (*Chen et al., 2013*; *Shern et al., 2014*) and 82.5% of RMS tumors have been found to display a strong activation of the pathway (*Renshaw et al., 2013*). LY 294002 (herein referred as LY), a PI3K pathway inhibitor (*Vlahos et al., 1994*), was employed in our panel of murine ERMS cells. In two cell lines (#59 and #2320), LY dramatically reduced cell proliferation (*Figure 6E,F*), strongly inhibited Akt phosphorylation, promoted accumulation in the G0/G1 phase of the cell cycle and induced apoptosis (*Figure 6—figure supplement 1E–G*). To investigate whether the addiction to Met and PI3K were mutually exclusive, PHA-sensitive cells (#1640 and #1796) were treated with LY, while LY-sensitive cells (#59 and #2320) were grown in the presence of PHA. PHA and LY did not interfere, respectively, with the proliferation rate of #59/# 2320 and #1640/#1796, confirming that these ERMS cells depend on the sustained activation of distinct signaling pathways (*Figure 6A, B, E, F*).

## Treatments based on combination therapy can be effective in preventing ERMS recurrence and clonal evolution

Unfortunately, almost all cancer patients treated with a drug aimed at a single target relapse and this is particularly evident in cases of intratumoral heterogeneity, in which such treatment inevitably results in the selection and expansion of resistant clones. In our effort to stabilize murine ERMS cell lines for pre-clinical studies, we routinely implanted the tumor in the left (L) and right (R) flanks of recipient nude mice. Intriguingly, in the case of tumor #187 we obtained two cell lines (#187L and R), one of which (R) expressed high levels of phosphorylated Met while the other one (L) did not (*Figure 7A*). R cells, as expected, were sensitive to PHA and Crizotinib (*Figure 7C, D, F, G, Figure 7—figure supplement 1A–C*). Conversely, L cells were resistant to PHA but responded to Crizotinib (*Figure 7D*, *Figure 7—figure supplement 1A*), suggesting amplification of *Alk*, a lesion frequently found in human RMS (*van Gaal et al., 2012*). Indeed, while visualization of ALK protein by Western blot in the primary tumor extract required enrichment by immunoprecipitation (*Figure 7B*), L cells expressed unusually high levels of ALK at baseline (*Figure 7A*). In L cells Crizotinib caused a strong inhibition of ALK signaling, impairment of anchorage-independent growth and induction of myogenic differentiation (*Figure 7E–G*), as well as G0/G1 accumulation and induction of apoptosis (*Figure 7—figure supplement 1B,C*). Accordingly, CGH and CNV analysis revealed

*Met* and *Alk* amplification, respectively, in R and L cells (*Figure 7H*, *Figure 7—figure supplement 1D*). Interestingly, real-time PCR showed a low level of *Alk* expression also in R cells (*Figure 7—figure supplement 1E*), suggesting that the original tumor contained a mixed population with a minor *Alk*-amplified component. We therefore treated R cells for one month with 250nM PHA, a dose that specifically inhibits Met signaling without interfering with the ALK pathway (*Figure 6—figure supplement 1A*, *Figure 7—figure supplement 1F,G*). PHA-selected R cells were negative for phospho-Met (*Figure 7I*), but were sensitive to Crizotinib (*Figure 7J*). CGH, CNV and real-time PCR analysis revealed the expansion, upon PHA treatment, of the *Alk*-amplified population (*Figure 7H*, *Figure 7—figure supplement 1D,E*). Conversely, prolonged PHA treatment of #1640 cells resulted in the emergence of a clone harboring also *Kras* amplification (*Figure 5E*, *Figure 5—figure supplement 3C*). Notably, the combination of Met and MEK inhibitors resulted in complete arrest of proliferation in these cells (*Figure 7K*). Overall these data suggest the presence of intratumoral heterogeneity in our model and that combination therapy can be a more effective strategy in preventing recurrence due to clonal evolution in ERMS.

## Discussion

ERMS and UPS are two distinct subtypes of sarcoma. While ERMS is characterized by features of myogenic cells, UPS lacks any tissue-specific marker of differentiation. Intriguingly, although phenotypically and genetically distinguishable, ERMS and UPS have emerged as parts of a tumor continuum, suggesting an intrinsic relationship between these two subtypes (*Blum et al., 2013*; *Rubin et al., 2011*). Our muscle-restricted inducible HGF model is a unique tool to explore the influence of SC niche perturbation (an issue which has not been thoroughly investigated so far) in the development of these two distinct but associated pathological entities. Interestingly, in a *Pax7* wild type background, where satellite cells are present in normal number, HGF induction prevalently resulted in ERMS formation with only less than 10% of mice developing UPS. Satellite cells are indeed particularly sensitive to the activation of the HGF/Met axis. They respond in vitro to HGF stimulation with an increase in proliferation rate (*Allen et al., 1995*), and injection of HGF in uninjured muscle directly stimulates satellite cells activation (*Tatsumi et al., 1998*). Recently, it has been shown that quiescent satellite cells residing in the controlateral limb with respect to the site of an injury respond to distant tissue damage by transitioning into an 'alert' state, a process that is again dependent on the HGF/Met axis (*Rodgers et al., 2014*). In the absence of satellite cells, muscle regeneration is dramatically impaired and this results in the aberrant deposition of fibrotic tissue (*Maltzahn et al., 2013*; *Murphy et al., 2011*; *Oustanina et al., 2004*). This process depends on the extensive expansion of resident fibroblasts, the other major component of the niche. Our model of perturbation of the SC niche showed that in a *Pax7* null genetic background ERMS incidence dramatically drops in favor of UPS development. In this background, while rare ERMS may still derive from the few remaining satellite cells (*Oustanina et al., 2004*), UPS are likely to originate from the aberrant expansion of fibroblasts (*Maltzahn et al., 2013*; *Murphy et al., 2011*). Accordingly, while murine ERMS were positive for satellite cell markers, UPS were positive for mesenchymal markers, such as PDGFRα. Interestingly, PDGFRα expression has been observed in 62% of human UPS samples (*Rüping et al., 2014*). Furthermore, bioinformatic analysis on human ERMS and UPS datasets revealed a satellite signature for ERMS and a fibroblast signature for UPS. Thus, changes in the microenvironment of the niche produce distinct subtypes of sarcoma depending on different susceptible cell types.

Genetic dissection of the cell of origin of ERMS has been extensively investigated. The preferred line of action has been the introduction of the most frequent human genetic lesions in mouse models by using cell/lineage specific promoters. However, these efforts, rather than unequivocally identifying the cell of origin, have further highlighted the complexity of sarcomagenesis. Hatley et al. (*Hatley et al., 2012*) obtained an aggressive form of head and neck ERMS by activating the Sonic Hedgehog pathway in combination with *Cdkn2a* deletion in adipocyte precursors. According to Rubin et al. (*Rubin et al., 2011*), 100% of ERMS incidence occurred when *p53* was knocked out in late myogenic precursors (*Myf6*-driven Cre), while its deletion in satellite cells (*Pax7*-driven Cre) resulted in UPS. Notably, activation of Sonic Hedgehog in satellite cells in the same context of *p53* deficiency, triggered ERMS formation, while its dysregulation in *Myf6*-positive-cells, promoted UPS development. Conversely ERMS, rather than UPS, represented the major histological subtype when

*Pax7*-driven *Kras*^G12D was combined with loss of *Trp53* (*Van Mater et al., 2015*). In this genetic setting, tumorigenesis was accelerated upon muscle injury or local administration of HGF. Notwithstanding, the same group showed that *MyoD*-driven *Kras*^G12D in association with loss of *Trp53* principally resulted in UPS rather than ERMS. Furthermore, expression of the activated form of the Hippo effector *YAP1* in satellite cells resulted in fully penetrant ERMS only upon injury (*Tremblay et al., 2014*). These models suggest two important mechanistic concepts. First, a given sarcoma subtype likely arises from the combination of specific genetic lesions with a well-defined cell lineage/stage of maturation. Second, tissue damage, such as injury or trauma, accelerates or, as in the case of YAP, triggers sarcoma development when occurring in a genetically predisposed background. Instead of imposing the most frequent genetic lesions to a cell in a specific developmental stage, we aimed at understanding how the most relevant component of the stem niche microenvironment, HGF, can contribute to ERMS and UPS formation. With our model, where only the soil of the niche is specifically modified while cell-intrinsic genetic events are left to chance, we proved that the HGF/Met axis mediates satellite cells release from quiescence, promoting their activation. In a wild type background, HGF stimulation perturbed muscle stem cell homeostasis only slightly. Conversely, in a tumor prone background, HGF caused massive expansion of cycling satellite cells, triggering ERMS initiation. Although we could not determine at which stage of the myogenic lineage full blown transformation occurs, the histological presentation supports a model of multistep ERMS progression. In the first stages of the disease, HGF-mediated stem cell niche perturbation resulted in neomyogenesis. At this early stage, Pax7-positive cells surrounded old fibers and were still able to form novel small muscle fibers. However at later stages, upon several cycles of proliferation and most likely through the acquisition of additional genetic lesion(s), stem cells lost the ability to differentiate and formed full blown tumors. Genetic validation of satellite cells as ERMS initiators was obtained by finding a drastic reduction of their incidence in the *Pax7* null background.

In our model, activation of the HGF/Met axis is involved in the initiation of all tumors. While variable Met expression was present in all murine ERMS, its phosphorylation was observed only in *Met*-amplified tumors that were sensitive to Met inhibition. Thus in our system its role in maintenance seems to be limited only to tumors where the *Met* locus is amplified, presumably as a consequence of an additional genetic hit. The importance of the HGF/Met axis in ERMS tumorigenesis had been first shown by Sharp et al. by combining ectopic expression of the *Hgf* transgene with loss of the *Cdkn2a* locus (*Sharp et al., 2002*). In their model phosphorylated Met was detectable in most tumor samples, indicating widespread constitutive activation. This discrepancy between the two models is likely to be due to differences in the strength and tissue specificity of the promoters. In Sharp et al. transgenic *Hgf* was expressed by the tumors, while in our model this was not the case.

In human cancer *MET* is rarely mutated or amplified while its overexpression is frequently observed in a variety of tumors including RMS and UPS (*Ferracini et al., 1996*; *Lahat et al., 2011*; *Taulli et al., 2006*). Our bioinformatic analysis revealed an enriched Met score in both human RMS and UPS. New evidence indicates that Met expression in cancer cells is a paradigm of 'inherence' (*Boccaccio and Comoglio, 2013*). The innate presence of Met in tumor cells is attributed to cancer stem cells (CSC) that inherit Met expression from their physiological counterpart (normal stem/progenitor cells) as part of their normal phenotype. In the CSC context, Met is not only a marker of the stemness status of the tumor but supports the self-renewal program and the expansion of CSC. Indeed, the HGF/Met axis sustains the stem cell phenotype in glioblastoma and colon cancer. Colon-derived xenospheres express Met in stem cell medium, but its downregulation rapidly occurs in differentiating conditions (*Luraghi et al., 2014*). In RMS as well, the Met pathway could sustain a stem cell phenotype. Our bioinformatic analysis on human datasets suggests that the Met and satellite signatures converge on the ERMS subtype, and our in vivo data clearly show that the HGF/Met axis acts prevalently on muscle stem cells to promote ERMS initiation. Furthermore, we previously showed that sustained expression of Met in RMS is partly due to the loss of muscle-specific microRNAs (*Taulli et al., 2009*). Ectopic introduction of myomiRs in RMS cells re-activated the differentiation program, in a process that involved Met downregulation and epigenetic reprogramming (*Coda et al., 2015*; *Taulli et al., 2014*), again confirming the role of Met in maintaining an undifferentiated, stem cell-like phenotype. However, the functional significance of Met in cancer has also been attributed to another property of some tumor cells, Met-'addiction'. Tumor cells harboring *MET* amplification display exquisite sensitivity to Met inhibitors, providing a rationale for the use of targeted therapies in patients carrying this lesion (*Smolen et al., 2006*). Accordingly, in our model

ERMS tumors bearing *Met* amplification showed Met-addiction. In these cases, Met inhibition stopped tumor growth and promoted terminal differentiation of the rare surviving cells, suggesting that, while the tumor bulk mainly consisted of Met-addicted cells, some residual cells required Met activity to retain their stem cell phenotype. These results nicely illustrate the two distinct roles of HGF/Met signaling in tumorigenesis based, respectively, on Met activity and Met-addiction and altogether suggest that, in the rare cases of human ERMS harboring this lesion, the patient may benefit from the use of a Met inhibitor.

Recent deep sequencing data of multiple regions of human tumors have shown that they often harbor a dominant genetic clone, plus one or more genetically and topologically distinct subclones (*Gerlinger et al., 2012*). The latter represent a major roadblock for a targeted therapeutic approach in terms of their potential to foster both progression and resistance (*McGranahan and Swanton, 2015*). Intratumoral heterogeneity is poorly reproduced in experimental models. Importantly, our model also exemplifies this feature, as shown by the isolation, from the same tumor, of a major *Met-* and a minor emerging *Alk-* or *Kras-*driven population. Expansion of the latter occurred following treatment with a Met inhibitor, underscoring the risk of drug-induced clonal evolution. Overall, our model demonstrates the functional relevance of the SC niche in ERMS and UPS development, providing an alternative explanation for the existence of a tumor continuum between these two subtypes. Although the high level of genetic complexity of sarcomas cannot be fully resolved with our model, we provide a rationale for the use of combination therapy instead of a single agent for a robust precision-guided therapeutic approach of genetically heterogeneous sarcomas.

## Materials and methods

All reagents, unless specified, are from Sigma-Aldrich (St. Louis, MO).

### Transgenic mice

*Ckm*-tTA mice were donated by P. Plotz (*Nagaraju et al., 2000*). The *Hgf* cDNA was inserted in the PvuII site of pBI-GFP vector (Clontech, Mountain View, CA). The fragment obtained after AseI digestion was microinjected into FVB fertilized eggs at the San Raffaele-Telethon Core Facility for Conditional Mutagenesis (Milano, Italy). *Cdkn2a*-null mice were from Jackson Laboratory (Bar Harbor, ME). *Pax7*-null mice were provided by P. Gruss. Genotyping primers are specified in *Supplementary file 1*. Doxycycline diluted in drinking water (1 mg/ml) was changed every 3 days. Approval for all animal procedures was granted by the Ethical Committee of the University of Torino during the session held in Turin on Sept 26, 2013 and later communicated to the Italian Ministry of Health on Oct 24, 2013.

### Semi-quantitative reverse transcription PCR and real-time PCR

RNA was extracted using TRIzol (Invitrogen, Carlsbad, CA) and retrotranscribed to cDNA using the iScript cDNA Synthesis Kit (Bio-Rad, Hercules, CA). Real-time PCR was performed with iQ SYBR Green (Bio-Rad). Primers are specified in *Supplementary file 1*.

### ELISA

Disaggregated *Tibialis anterior* muscles and sera were lysed in lysis buffer [20 mmol/L Tris (pH 7.5), 150 mmol/L NaCl, 1 mmol/L EDTA,1 mmol/L EGTA, 1% Triton X-100, 1 mmol/L h-glycerolphosphate] with Protease Inhibitor Cocktail. 200 μg of proteins were used for HGF determination using ELISA Kit (B-Bridge, Santa Clara, CA) according to the manufacturer's protocol.

### Immunohistochemistry and cross-sectional area analysis

Immunohistochemistry was performed as previously described (*Taulli et al., 2009*) with the specified antibodies. Ki67 (#NCL-Ki67p) was from Leica Biosystems (UK); MyoD (#M3512) was from Dako (Denmark); Myogenin (DSHB (Iowa City, IA) Hybridoma Product F5D was deposited by Wright, Woodring E.); Pax7 (DSHB Hybridoma Product PAX7 was deposited by Kawakami, Atsushi); eMHC (DSHB Hybridoma Product F1.652 was deposited by Blau, H.M.); P-Met (#3126) and PDGFRα (#3164) were from Cell Signaling Technology (Danvers, MA); Met (#18–7366) was from Invitrogen.

Fiber cross-sectional areas were measured using ImageJ software (rsb.info.nih.gov/ij). Statistical analyses were performed using Microsoft Excel software, with a moving average (period 4) trendline.

## Muscle sections staining, single fiber isolation, culture and immunofluorescence

Muscles were immersed into isopentane for 30 sec and frozen in liquid nitrogen. Cryosections were fixed with 4% paraformaldehyde, permeabilized with methanol at -20°C for 6 min and incubated with the specified antibodies. Laminin (#L9393) and Desmin (#D1033) were from Sigma-Aldrich; Pax7 (DSHB Hybridoma Product PAX7 was deposited by Kawakami, Atsushi); MyoD (#sc-760) was from Santa Cruz Biotechnology (Dallas, TX); 488, 555 and Cy3-conjugated secondary antibodies were from Invitrogen. For Pax7 and MyoD the signal was amplified by incubation with biotin-conjugated goat anti-mouse/rabbit IgG1 (Jackson ImmunoResearch, West Grove, PA) followed by incubation with 488-coniugated streptavidin (Jackson ImmunoResearch) or with Cy3-coniugated streptavidin (Jackson ImmunoResearch). Nuclei were stained with DAPI. EDL single fiber isolation was performed as previously described (*Zammit et al., 2002*). Single fibers were plated in matrigel-coated culture plates and maintained in DMEM supplemented with 0.5% Chicken Embryo extract, 10% Horse Serum. Fibers were fixed in 4% PAF and permeabilized in 0.5% Triton X-100. Nonspecific antibody binding was blocked by incubation in 20% PBS-BSA. Fibers were incubated overnight at +4°C with the specified antibodies. Pax7 (DSHB Hybridoma Product PAX7 was deposited to the DSHB by Kawakami, Atsushi); MyoD (#sc-760) was from Santa Cruz Biotechnology. Alexa 555-conjugated secondary antibodies were from Invitrogen. Nuclei were stained with DAPI. The number of cells/fiber was counted ( ± SEM).

## Cell culture and inhibitors

Murine ERMS cells and satellite cells were isolated as previously described (*Crepaldi et al., 2007*). ERMS cells were maintained in DMEM supplemented with 10% FBS. Satellite cells were maintained as previously described (*Crepaldi et al., 2007*). SU-DHL-1 and TS cells (a subclone of SUP-M2) were kindly provided by R. Chiarle and maintained in RPMI 1640 supplemented with 10% FBS. Methods of characterization from the cell bank include karyotyping and DNA fingerprinting. All cell lines were regularly tested with MycoAlert (Lonza, Walkersville, MD) to ascertain that cells were not infected with mycoplasma. Met inhibitor PHA-665752 was used at 250 nmol/L. Crizotinib (PF-02341066) (Selleckchem, Houston, TX) was used at 250 nmol/L. LY-294002 was used at 10 µmol/L. Selumetinib (AZD6244) (Selleckchem) was used at 1 µmol/L. DMSO was used as a control.

## Western blot, Immunoprecipitation and Phospho RTK-Array

Western blot was performed using the specified antibodies. MHC (#sc-32732) and Myogenin (#sc-12732) were from Santa Cruz Biotechnology; Met (#18–7366) was from Invitrogen; P-Met (#3126), P-Akt (#9271), GAPDH (#5174), P-TYR (#9411), hALK (#3633), P-hALK (#3341) and PDGFRα (#3164) were from Cell Signaling Technology; mALK (#ab16670) was from Abcam (UK); MyoD (#M3512) was from Dako; P-MAPK (# M8159), Tubulin (#T5201), Actin (#A5060) were from Sigma-Aldrich. mALK immunoprecipitation: total cell extracts (RIPA) were 5x diluted in IP buffer (50 mM TrisHCl pH 7.9; 150 mM NaCl; 1 mM EDTA; 5 mM MgCl2; 0.1% NP-40; 20% glycerol) with 1 mM phenylmethylsulfonyl fluoride, 10 mM NaF, 1 mM Na3VO4 and protease inhibitor cocktail. Samples were precleared with equilibrated Dynabeads protein G (Invitrogen); mALK antibody and normal-IgG (Santa Cruz Biotechnology) were incubated overnight at 4°C. Mouse Phospho-Receptor Tyrosine Kinase Array (R&D Systems, Minneapolis, MN) was performed following the manufacturer's instructions. The mean pixel density was measured using Quantity One software and expressed as a percentage of the mean pixel density of positive controls.

## Cell proliferation assay, cell-cycle analysis and assessment of apoptosis

Cells were seeded in 12-well plates at a density of $1 \times 10^4$ cells/well. Proliferation was evaluated by CellTiter-Glo (Promega, Madison, WI) following the manufacturer's instructions. Cell-cycle and apoptosis analyses were performed as previously described (*Taulli et al., 2009*).

## Membrane staining for FACS analysis

Cells were washed in PBS and incubated for 30 min with primary antibodies (anti Sca1: FITC Rat Anti-Mouse Ly-6A/E BD Biosciences, Franklin Lakes, NJ, #553335; anti mouse Integrin-α7 MBL International, Woburn, MA, #K0046-3) diluted in PBS-0.1% BSA. After washing, cells were incubated for 30 min with secondary antibody (required only for Integrin-α7: APC goat anti-mouse IgG Invitrogen #A865) and resuspended in PBS-0.1% BSA. Cells were analyzed by FACS scan using CellQuest Software (BD Biosciences).

## MHC staining

MHC immunofluorescence and MHC staining for FACS analysis was performed as previously described (*Coda et al., 2015*; *Taulli et al., 2009*).

## Anchorage-independent cell growth assay

Cells were suspended in 0.45% type VII low-melting agarose in 10% FBS DMEM at $5 \times 10^4$/well and plated on a layer of 0.9% agarose in 10% FBS DMEM in 6-well plates and cultured for two weeks at 37°C with 5% $CO_2$.

## In vivo tumorigenesis assay

Cells were trypsinized and resuspended at $4 \times 10^6$ cells/ml in sterile PBS. Cells (100 μl) were injected subcutaneously into the flank of female *nu/nu* mice (Charles River Laboratories, Wilmington, MA). Tumor size was measured with Vernier calipers every 2 days and tumor volumes were calculated as a sphere volume. Mice were treated for 8 days with 100 mg/kg/day Crizotinib by oral gavage. Crizotinib was resuspended in 0.5% methylcellulose and 0.4% polysorbate 80.

## CGH and CNV analysis

Comparative genomic hybridization analysis was performed at 'Fondazione Edo ed Elvo Tempia', Biella, Italy. Total DNA was extracted using the DNeasy Blood & Tissue Kit (Qiagen, Germany). Comparative genomic hybridization using aCGH microarrays wad carried out using the enzymatic labeling method. Digestion, labeling, hybridization, washing and slide scanning were performed following the manufacturer's protocols (Agilent Technologies, Santa Clara, CA). Images were analyzed using Feature Extraction software version 10.7 (Agilent Technologies). CNV primers are specified in *Supplementary file 1*. Analysis was performed using the standard curve method. *Actl6a* was used as a control: for each sample, the copy number was calculated using the ratio of *Met/Kras/Alk* Vs *Actl6a* copy number. Relative copy number variation was determined for each sample in comparison with a non-amplified reference sample.

## RNAseq data

RNA sequencing of primary tumors was performed as previously described (*Shern et al., 2014*). Briefly, PolyA-selected RNA libraries were prepared for RNA sequencing on Illumina HiSeq2000 and Illumina NextSeq using TruSeq v3 chemistry according to the manufacturer's protocol (Illumina, San Diego, CA). One hundred base-long paired-end reads were assessed for quality and reads were mapped using CASAVA (Illumina). The generated fastq files were used as input for TopHat2 (*Trapnell et al., 2009*). Reads were mapped according to the hg19 human genome assembly. Cufflinks (http://cufflinks.cbcb.umd.edu/) (*Trapnell et al., 2010*) was used to assemble and estimate the relative abundances of transcripts mapped with TopHat2 at the gene and transcript level (FPKM). FPKM values were log2 transformed. Sample specific Z-scores of expression were calculated using a panel of normal tissue. The generated data is publically available via the National Cancer Institute Oncogenomics web site (http://pob.abcc.ncifcrf.gov/cgi-bin/JK).

## FISH

Formalin-fixed paraffin-embedded tissue sample was collected at Fondazione IRCCS Istituto Tumori in Milan, Italy, according to the Internal Review and the Ethics Boards of the Istituto Nazionale dei Tumori of Milan. Patients gave their written consent for research activities. FISH was performed on 2–4 μm-thick paraffin embedded sections by counting at least 100 tumor cells. *MET* amplification was assessed using a commercial probe at 7q31 (Zyto-Light SPEC MET/CEN7 Dual Color Probe,

ZytoVision, Germany), used according to the manufacturer's instructions. MET/CEN7 ratio, the percentage of tumor cells with >4 MET signals, and the average MET copy number per cell were calculated. FISH results were evaluated in concordance with the criteria described by Schildhaus et al. (*Schildhaus et al., 2015*).

## Human gene expression data analysis and signature determination

We ran expression analysis on the following publically available sets of microarray data: for 'Davicioni' dataset (Affymetrix Human Genome U133A Array platform) we used 134 RMS (*Davicioni et al., 2009*). For 'Williamson' dataset (Affymetrix Human Genome U133 Plus 2.0 Array platform) we used 101 RMS samples (*Williamson et al., 2010*). Muscle datasets GSE3307 and GSE1462 were used in comparison with RMS dataset 'Davicioni', while muscle datasets GSE3526 and GSE2328 were used in comparison with RMS dataset 'Williamson'. For 'Gibault' dataset (Affymetrix Human Genome U133 Plus 2.0 Array platform) we used 73 UPS samples and 79 other sarcoma samples (20 myxofibrosarcomas MFS, 10 liposarcomas LPS and 49 leiomyosarcomas LMS) (*Gibault et al., 2011*).

All samples were normalized by the RMA algorithm as implemented in R free software environment, with custom CDF (*Lembo et al., 2012*). Differential expression was evaluated by limma package.

As satellite signature we used the overlap between the mouse signatures as in Fukada (*Fukada et al., 2007*) and in Pallafacchina (*Pallafacchina et al., 2010*) translated in human genes by the Homologene mapping, build 67. Fukada's gene IDs have been mapped to mouse Entrez IDs by RefSeq database, release 57, and Pallafacchina's gene IDs by affymetrix mouse430_2 annotation, release na31. As Met signature, we used Bertotti signature (*Bertotti et al., 2009*). Clustering classification was made on the respective signature genes found on the platform by Bioconductor *hclust* function. As fibroblast signature, we derived differentially expressed genes between H9F cardiac fibroblasts cell line (n=3 samples) and H9 cardiomyocytes cell line (n=3 samples) by using GEO2R (http://www.ncbi.nlm.nih.gov/geo/geo2r/), imposing a cut-off of 0.05 on the adjusted p-value on the selection of differentially expressed genes (*Fu et al., 2013*). Signatures scores are essentially the algebraic sum of the signature gene expression level: the contribution is added if the gene is up in the signature, subtracted otherwise.

## Statistical analysis

2-tailed unpaired Student's $t$ test was used to evaluate statistical significance: [NS]$p>0.05$; *$p<0.05$; **$p<0.01$; ***$p<0.001$; ****$p<0.0001$. In the CGH analysis, raw data were processed using the Agilent Genomic Workbench version 7. Aberrant regions were detected using ADM-2 algorithm with threshold set to 6. To avoid false positive calls, the minimum number of consecutive probes for amplifications/deletions was set at 3, together with a minimum average absolute Log Ratio for aberrations $\geq 0.25$. For human datasets analysis gene differential expression was based on the bioconductor limma package. Enrichments have been evalutated by hypergeometric test. Heteroscedastic one-tailed Student t-test has been used to evaluate the differential signature's scores.

## Acknowledgements

The authors wish to thank Dr. Maria Stella Scalzo (University of Torino) for help with immunohistochemical analysis. The support of the Fondazione Ricerca Molinette Onlus is gratefully acknowledged.

## Additional information

### Funding

| Funder | Grant reference number | Author |
|---|---|---|
| Italian Association for Cancer Research | AIRC Project number, IG12089 | Carola Ponzetto |
| Regione Piemonte | IMMONC Project | Carola Ponzetto |

| Italian Association for Cancer Research | AIRC Start-Up, Project number 15405 | Riccardo Taulli |
|---|---|---|
| Associazione Bianca Garavaglia | A/15/01N | Gabriella Sozzi |

The funders had no role in study design, data collection and interpretation, or the decision to submit the work for publication.

### Author contributions
DM, RT, Conception and design, Acquisition of data, Analysis and interpretation of data, Drafting or revising the article; NM, FB, PEF, MFL, VF, PŠ, SM, AM, JFS, JK, UA, PP, VS, TC, PG, MC, AF, Acquisition of data, Analysis and interpretation of data; GS, RC, Analysis and interpretation of data, Drafting or revising the article; CP, Conception and design, Analysis and interpretation of data, Drafting or revising the article

### Author ORCIDs
Valentina Sala, http://orcid.org/0000-0002-2283-2807
Riccardo Taulli, http://orcid.org/0000-0003-1277-6263

### Ethics
Human subjects: Formalin-fixed paraffin-embedded tissue sample was collected at Fondazione IRCCS Istituto Tumori in Milan, Italy, according to the Internal Review and the Ethics Boards of the Istituto Nazionale dei Tumori of Milan. Patients gave their written consent for research activities.
Animal experimentation: Approval for all animal procedures was granted by the Ethical Committee of the University of Torino during the session held in Turin on Sept 26, 2013 and later communicated to the Italian Ministry of Health on Oct 24, 2013.

## Additional files

### Supplementary files
• Supplementary file 1. List of primers used for genotyping, semi-quantitative reverse transcription PCR, real-time PCR primers and CNV analysis.

### Major datasets
The following previously published datasets were used:

| Author(s) | Year | Dataset title | Dataset URL | Database, license, and accessibility information |
|---|---|---|---|---|
| Hoffman E | 2005 | Comparative profiling in 13 muscle disease groups | http://www.ncbi.nlm.nih.gov/geo/query/acc.cgi?acc=GSE3307 | Publicly available |
| Crimi M, Comi GP | 2005 | Mitochondrial disorders | http://www.ncbi.nlm.nih.gov/geo/query/acc.cgi?acc=GSE1462 | Publicly available |
| Davicioni E | 2006 | Identification of a PAX-FKHR gene expression signature that defines molecular classes and determines the prognosis of alveolar rhabdomyosarcomas | ftp://caftpd.nci.nih.gov/pub/caARRAY/experiments/caArray_trich-00099/ | Publicly available |
| Roth RB | 2006 | Comparison of gene expression profiles across the normal human body | http://www.ncbi.nlm.nih.gov/geo/query/acc.cgi?acc=GSE3526 | Publicly available |

| | | | | |
|---|---|---|---|---|
| Cobb JP, Mindrinos MN, Miller-Graziano C, Calvano SE, Baker HV, Xiao W, Laudanski K, Brownstein BH, Elson C, Hayden DL, Herndon D, Lowry SF, Maier RV, Schoenfeld D, Moldawer LL, Davis R, Tompkins RG | 2005 | Application of genome-wide expression analysis to human health & disease | http://www.ncbi.nlm.nih.gov/geo/query/acc.cgi?acc=GSE2328 | Publicly available |
| Williamson D | 2011 | E-TABM-1202 - Transcriptional profiling by array of primary rhabdomyosarcoma samples with different PAX3/FOXO1 fusion gene status | http://www.ebi.ac.uk/arrayexpress/files/E-TABM-1202/ | Publically available |
| Gibault L, Aurias A | 2010 | Expression data from human soft tissue sarcomas with complex genomics | http://www.ncbi.nlm.nih.gov/geo/query/acc.cgi?acc=GSE23980 | Publicly available |
| Fukada S | 2007 | Genome-wide expression analysis of satellite cells | http://www.ncbi.nlm.nih.gov/geo/query/acc.cgi?acc=GSE3483 | Publically available |
| Pallafacchina G, Montarras D, Regnault B, Buckingham M | 2010 | Gene profiling of quiescent and activated skeletal muscle satellite cells by an in vivo approach | http://www.ncbi.nlm.nih.gov/geo/query/acc.cgi?acc=GSE15155 | Publicly available |
| Medico E, Trusolino L, Bertotti A, 2010 | 2010 | Integrative 'omic' approaches identify a Ras/PI3K signature sustaining addiction to the MET oncogene | http://www.ncbi.nlm.nih.gov/geo/query/acc.cgi?acc=GSE19043 | Publicly available |
| Fu J, Quan L | 2013 | Cell Reprogramming experiment (reprogramming cardiac fibroblasts into cardiomyocytes) | http://www.ncbi.nlm.nih.gov/geo/query/acc.cgi?acc=GSE49192 | Publicly available |

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
