## [Decision Letter]

Thank you for submitting your work entitled "HGF-mediated satellite cells niche perturbation promotes development of distinct sarcoma subtypes" for consideration by *eLife*. Your article has been favourably assessed by Fiona Watt (Senior Editor), a Reviewing Editor, and three reviewers. Two of the three reviewers have agreed to reveal their identity: Beat Schafer and Yanlin Yu.

The reviewers have discussed the reviews with one another and the Reviewing Editor has drafted this decision to help you prepare a revised submission.

Embryonal rhabdomyosarcomas (ERMS) and undifferentiated pleomorphic sarcomas (UPS) that develop in childhood and adults, respectively, are considered to represent a continuum of tumors from previous mouse model studies. Morena et al. describe the development of further mouse models that are Pax-7 dependent and shed light on the impact of HGF expression in the tumor microenvironment on the initiation of these sarcoma types. This represents a unique model for ERMS and UPS and the genetic heterogeneity and dependencies on pathways in tumors is shown to influence drug sensitivities.

This is an interesting and important study and the experiments are well executed, however the following points need to be addressed:

1) To gain more mechanistic insights, primary cell cultures were established from ERMS, however this was not possible from UPS carrying mice. Using these cell lines, the authors demonstrate tumor heterogeneity in terms of activated signaling pathways and sensitivity to inhibitors. This is very intriguing, but it is unclear how this relates to heterogeneity in vivo or how much was induced by in vitro culturing conditions. This should be clarified to further strengthen the model by e.g. looking directly at heterogeneity in tumor sections. How does such heterogeneity relate to human tumors?

2) Expression analyses that justify the involvement of the MET pathway and a link to the potential cell of origin through a high satellite cell signature in ERMS are presented (Figure 1 etc.). However, no such evidence for involvement of MET is provided for UPS that would suggest that the model is representative of the human UPS.

3) The authors suggest fibroblasts or myofibroblasts as a cell of origin for UPS – is there any evidence in the analyses of expression data to support this in addition to the cell's presence?

Markers of fibroblasts should be examined in UPS to check for aberrant expansion of this cell population.

4) In a previous study (Sharp et al., 2002) it was shown that HGF/Met signaling was constitutively activated in most of RMS derived from HGF transgenic with *Ink4/Arf* deficient mice. The authors should give an explanation of the different activation of HGF/Met signaling in RMS in a similar mouse model.

The authors propose that HGF/Met signaling functioned at early stages of ERMS development. Could the authors examine the status of activated Met from early to late stages of ERMS in the mouse model?

5) Can the authors reconcile the strong Met signature identified in human tumors with the HGF/Met independent growth of tumors in the models?

6) Justification should be given for the use of *Ink4a/Arf* null mice as representative of the human sarcomas.

7) The cell of origin of RMS is still debated as stated in paragraph one, Introduction. How this study supports the satellite PAX7 expressing cell as the cell of origin could be further developed in the Discussion.

---

## [Author Response]

[…] This is an interesting and important study and the experiments are well executed, however the following points need to be addressed:

*1) To gain more mechanistic insights, primary cell cultures were established from ERMS, however this was not possible from UPS carrying mice. Using these cell lines, the authors demonstrate tumor heterogeneity in terms of activated signaling pathways and sensitivity to inhibitors. This is very intriguing, but it is unclear how this relates to heterogeneity in vivo or how much was induced by in vitro culturing conditions. This should be clarified to further strengthen the model by e.g. looking directly at heterogeneity in tumor sections. How does such heterogeneity relate to human tumors?*

We understand the concern of the referees regarding the possibility that heterogeneity of genetic lesions may be a result of culturing, rather than reflecting the features of the original tumor.

With respect to Met as a driver, Western blot and immunohistochemical analysis of sections of primary tumors (Figure 5 and new Figure 5) show strong *Met* expression and activation only in tumors which yielded cell lines subsequently found *Met*-amplified (Figure 5, Figure 5—figure supplement 3) and Met addicted (Figure 6, Figure 6—figure supplement 1). These findings have been commented in the revised version of the manuscript (paragraph two, subheading “SC niche perturbation results in heterogeneous tumors, with only a subset displaying sensitivity to Met or PI3K pathway inhibition”). With regards to how such Met heterogeneity is related to human tumors, a genomic analysis of a large panel of human RMS (134 cases) revealed *MET* amplification in only one sample (Figure 5). In our manuscript, we report another ERMS with *MET* amplification that emerged from FISH analysis of a much smaller cohort of human ERMS (8 cases) (Figure 5). A comment regarding the consistency between murine and human data has been included in the revised version of the manuscript (paragraph three, subheading “SC niche perturbation results in heterogeneous tumors, with only a subset displaying sensitivity to Met or PI3K pathway inhibition”).

As far as intratumoral heterogeneity, which is important because it may be at the basis of resistance to targeted therapy, although we cannot offer proof of a mixed population of cells expressing different drivers in sections of the original tumors, we would like to point out what we believe is indirect evidence. In the case of tumor #187, we obtained two ERMS cell lines (L and R) driven – respectively – by ALK and Met, by transplantation of two specimens of the primary tumor in the two flanks of a nude mouse. Real-time PCR showed that a low level of *Alk* expression was already present in R cells, suggesting that the *Alk*-amplified clone, emerged from selection with a Met-specific inhibitor (PHA), was already present in the original tumor (subheading “Treatments based on combination therapy can be effective in preventing ERMS recurrence and clonal evolution”). This conclusion is reinforced by the low number of passages (less than 10 in only one month of selection) which were necessary to obtain the *Alk*-amplified clone. As we reported in the manuscript (subheading subheading “Treatments based on combination therapy can be effective in preventing ERMS recurrence and clonal evolution”), lesions in the *ALK* gene have been repeatedly found in human ERMS, again strengthening the congruity of our model.

*2) Expression analyses that justify the involvement of the MET pathway and a link to the potential cell of origin through a high satellite cell signature in ERMS are presented (Figure 1 etc.). However, no such evidence for involvement of MET is provided for UPS that would suggest that the model is representative of the human UPS.*

In the present version we have extended the analysis of the Met signature to a large dataset of human sarcomas (Gibault et al., 2011) comprising 73 UPS and 79 other sarcomas (20 Myxofibrosarcomas, 10 Liposarcomas, and 49 Leiomyosarcomas). The Met signature score was higher in UPS than in all the other subtypes. We included these results in new Figure 1, and modified the text accordingly (subheading “Met and satellite signatures are both preferentially associated with the ERMS subtype, while UPS show high Met and fibroblast scores.; paragraph four, Discussion). The new data, together with the paper of Lahat et al., (2011) describing activation of the Met pathway in human UPS, strengthen the relevance of our model (paragraph four, Discussion).

We have also included in new Figure 4 a panel showing that murine primary UPS express Met (subheading “SC niche perturbation in a Pax7-deficient background mainly results in UPS development”).

*3) The authors suggest fibroblasts or myofibroblasts as a cell of origin for UPS* – *is there any evidence in the analyses of expression data to support this in addition to the cell's presence?*

*Markers of fibroblasts should be examined in UPS to check for aberrant expansion of this cell population.*

As requested by the reviewers, we are now providing expression data (relative to both human UPS and to our model) to support the conclusion that fibroblasts may be the cell of origin of UPS.

We used a fibroblast signature for comparative analysis of human UPS and RMS. The fibroblast score was consistently higher in UPS samples. We included these results in new Figure 1 and modified the text accordingly (subheading “Met and satellite signatures are both preferentially associated with the ERMS subtype, while UPS show high Met and fibroblast scores.”; paragraph one, Discussion).

We analyzed primary murine UPS and RMS, for the presence of fibroblast markers (as described by Driskell et al., 2013; Guarnerio et al., 2015; Joe et al., 2010). Immunohistochemical and Western blot analysis showed a strong positivity for the pan fibroblast marker PDGFRα exclusively in UPS samples (new panels in Figure 4, new Figure 4). Flow cytometry analysis showed that murine UPS, in contrast to ERMS, were positive for the mesenchymal marker Sca1, and negative for the satellite cell marker Integrin-α7 (new Figure 4). Interestingly, PDGFRα expression has been observed in 62% of human UPS samples (Rüping et al., 2014). The text has been modified accordingly (subheading “SC niche perturbation in a Pax7-deficient background mainly results in UPS development”;paragraph one, Discussion).

*4) In a previous study (Sharp et al., 2002) it was shown that HGF/Met signaling was constitutively activated in most of RMS derived from HGF transgenic with Ink4/Arf deficient mice. The authors should give an explanation of the different activation of HGF/Met signaling in RMS in a similar mouse model.*

The two models use different promoters to express transgenic *Hgf*. In Sharp et al. *Hgf* was under the regulation of the strong metallothionein promoter, which is active in virtually all tissues. Thereby, the tumors themselves showed transgenic *Hgf* expression, which may explain sustained Met phosphorylation (Sharp et al.). In our model, *Hgf* expression is restricted to adult muscle, Met stimulation is paracrine and the ERMS tumors, given the lack of differentiated cells, no longer express transgenic *Hgf*. Furthermore, although all tumors retain Met expression (we have added in Figure 5 a longer exposure of the Met Western blot analysis), if Met is stimulated by endogenous HGF, its phosphorylation is below the limit of detection. We have described these findings in the Results section (paragraph one, subheading “SC niche perturbation results in heterogeneous tumors, with only a subset displaying sensitivity to Met or PI3K pathway inhibition”). In our model the few cases of murine ERMS with high levels of Met phosphorylation are linked to Met overexpression/amplification, a genetic event which occurs at low frequency also in human ERMS.

A comment regarding the different strategies used to generate transgenic HGF has been included in the Discussion of the revised version of the manuscript (paragraph three, Discussion).

*The authors propose that HGF/Met signaling functioned at early stages of ERMS development. Could the authors examine the status of activated Met from early to late stages of ERMS in the mouse model?*

In our model in a tumor prone background, HGF acts as the trigger of aberrant satellite cells activation and expansion. The fact that initially satellite cells were still able to differentiate into new fibers indicates that in our system the threshold of HGF and consequent activation of Met signaling is functionally significant, but relatively low. In fact Met can still be downregulated (new Figure 5—figure supplement 2), allowing differentiation to occur. At this stage we found it difficult to detect Met and P-Met by IHC (Figure 5—figure supplement 2). Additional genetic event(s) must be invoked to explain the block of myogenic differentiation that leads to progression. In a few cases one of these events consists of *Met* amplification. In *Met*-amplified tumors, where activation is most likely due to overexpression, Met and P-Met were easily detectable (Figure 5 and Figure 5—figure supplement 2). In all other tumors, at late stages basal Met is detectable (Figure 5 Met long exposure, Figure 5—figure supplement 2), but is not phosphorylated. The text has been modified to include these results (paragraph one, subheading “SC niche perturbation results in heterogeneous tumors, with only a subset displaying sensitivity to Met or PI3K pathway inhibition”)

*5) Can the authors reconcile the strong Met signature identified in human tumors with the HGF/Met independent growth of tumors in the models?*

In the revised version we have clarified the difference between Met signature (a model to measure the Met activity in each sample, definition now included in paragraph one, Results section) and Met-addiction (paragraph four, Discussion). A high “Met score” does not necessarily imply that the tumor is addicted to Met. Addiction to a driver is generally dictated by its genomic amplification (for *MET* see Smolen et al., 2006). Consistently, we show that only *Met*-amplified murine ERMS are sensitive to Met inhibitors. On the other hand, according to the concept of “inherence”, the Met signature of human ERMS could be inherited from the cell of origin. Its retention, rather than driving proliferation, may contribute to resistance to differentiation (Taulli et al., 2009) and proclivity to accumulate additional genetic hits. The Discussion has been revised to elucidate these two concepts (paragraph four, Discussion).

*6) Justification should be given for the use of Ink4a/Arf null mice as representative of the human sarcomas.*

Following the reviewers comment we have included a description of previous works demonstrating the relevance of the *CDKN2A* loss in sarcomagenesis (subheading “Loss of homeostatic balance between SC proliferation and differentiation results in a multistep model of ERMS development”).

*7) The cell of origin of RMS is still debated as stated in paragraph one, Introduction. How this study supports the satellite PAX7 expressing cell as the cell of origin could be further developed in the Discussion.*

As requested by the reviewers, we have further motivated the contribution of our model in support to the satellite cell origin of ERMS. The expanded explanation is now included in the revised form of the manuscript (paragraph two, Discussion).